# Positron Emission Tomography Radiopharmaceuticals in Differentiated Thyroid Cancer

**DOI:** 10.3390/molecules27154936

**Published:** 2022-08-03

**Authors:** Chaninart Sakulpisuti, Putthiporn Charoenphun, Wichana Chamroonrat

**Affiliations:** Division of Nuclear Medicine, Department of Diagnostic and Therapeutic Radiology, Faculty of Medicine Ramathibodi Hospital, Mahidol University, Bangkok 10400, Thailand; chaninart.sak@mahidol.ac.th (C.S.); putthiporn.cha@mahidol.ac.th (P.C.)

**Keywords:** positron emission tomography/computed tomography (PET/CT), PET radiopharmaceutical, differentiated thyroid cancer (DTC)

## Abstract

Differentiated thyroid cancer (DTC), arising from thyroid follicular epithelial cells, is the most common type of thyroid cancer. Despite the well-known utilization of radioiodine treatment in DTC, i.e., iodine-131, radioiodine imaging in DTC is typically performed with iodine-123 and iodine-131, with the current hybrid scanner performing single photon emission tomography/computed tomography (SPECT/CT). Positron emission tomography/computed tomography (PET/CT) provides superior visualization and quantification of functions at the molecular level; thus, lesion assessment can be improved compared to that of SPECT/CT. Various types of cancer, including radioiodine-refractory DTC, can be detected by 2-[^18^F]fluoro-2-deoxy-D-glucose ([^18^F]FDG), the most well-known and widely used PET radiopharmaceutical. Several other PET radiopharmaceuticals have been developed, although some are limited in availability despite their potential clinical utilizations. This article aims to summarize PET radiopharmaceuticals in DTC, focusing on molecular pathways and applications.

## 1. Introduction

Differentiated thyroid cancer (DTC), arising from thyroid follicular epithelial cells, constitutes the majority of thyroid malignancies (>90%) [1]. This category consists of papillary thyroid cancer (PTC), follicular thyroid cancer (FTC), and oncocytic carcinoma [2,3]. The incidence rates of DTC differ among countries; iodine inadequacy and access to healthcare are the two largest causes. In the United States, the incidence has progressively increased over the last decade, with an overall age-adjusted annual rate of 15/100,000 in 2015, and this resulted from the majority rise in incidental PTC [2]. Although the estimated 5-year disease-specific survival in PTC and FTC is excellent in patients with localized (99.9%) and regional (98.3%) disease, it is markedly lower in patients with distant metastasis (53.3%) [4]. Therefore, imaging plays an important role in diagnosis, treatment planning, and monitoring.

Ultrasonography, computed tomography (CT), and magnetic resonance imaging are anatomical imaging modalities recommended for the preoperative evaluation of the primary tumor and locoregional and distant metastases, and in searching for structural residual or recurrent disease in patients with high serum thyroglobulin (Tg) [1]. 

Nuclear medicine imaging, single photon emission tomography (SPECT), and positron emission tomography (PET) provide the visualization and quantification of functions at the cellular or molecular level. In combination with CT, lesion localization can be further defined [5]. For SPECT imaging, iodine-123 and iodine-131 (^131^I) are well-known types of radioiodine (RAI) used for detecting metastatic DTC after thyroidectomy because of the expression of the sodium-iodine symporter (NIS) [6]. PET is considered to have a better resolution and sensitivity compared to SPECT [5]**,** and several PET radiopharmaceuticals are currently accessible for clinical use. 

The aim of this review article is to summarize PET radiopharmaceuticals in DTC, focusing on molecular pathways and applications. 

## 2. Iodine-Related PET Radiopharmaceuticals

### Iodine-124 (^124^I) 

^124^I is a positron-emitting, long-lived PET radionuclide and has a physical half-life of 4.2 days [7]. It decays dual high-energy positrons (1532 keV (11%) and 2135 keV (11%)) and performs electron capture with gamma emissions of 511 keV (46 %), 603 keV (61 %), and 1691 keV (11 %) [7,8]. Although the reduced PET imaging resolution has been a concern due to ^124^I’s physical properties, namely (1) high energy positron emission with a long positron range before annihilation with an electron occurs, and (2) a high fraction of nonpositron decays, ^124^I PET imaging is feasible with current reconstruction algorithms [8,9].

^124^I, similarly to other RAI isotopes, is trapped by NIS in residual thyroid tissue or tumor in the thyroidectomy bed and metastatic lesion. Thyroid-stimulating hormone (TSH) is used to increase NIS expression [10]. Therefore, a low dose of ^124^I is orally administered after appropriate thyroid-stimulating hormone (TSH) stimulation, either by thyroid hormone withdrawal or two consecutive days of intramuscular injection of recombinant human TSH (rhTSH), along with a low-iodine diet. Subsequently, the ^124^I PET scan can be performed up to 96 h after tracer administration [11]. Wu et al. studied the optimal time for ^124^I PET/CT scan and suggested that dual-time-point imaging was superior to single-time-point imaging, and dual time points at 48 + 72 h or 48 + 96 h yielded the highest lesion detection rate [12].

Several studies performed ^124^I PET(/CT) for postoperative staging, pre-therapeutic dosimetry, follow-up treatment response, and assessment of persistent or recurrent disease [13,14,15,16,17,18,19,20,21,22]. Ruhlmann et al. found excellent agreement between pre-therapeutic ^124^I PET and post-therapeutic ^131^I whole-body scan (WBS) and SPECT/CT in detecting iodine-avid metastatic DTC [13], and Nostrand et al. demonstrated that ^124^I PET identified additional iodine-avid residual thyroid or metastases that were not seen with ^131^I [14]. A meta-analysis of five studies in DTC patients reported that ^124^I PET/CT was a highly sensitive imaging modality for detecting RAI-avid lesions amenable to RAI therapy (pooled sensitivity = 94.2%). It also detected some new lesions that were not visualized on the post-therapeutic ^131^I scan [23]. However, Khorjekar et al. found that negative ^124^I PET imaging had low predictive value for a negative post-therapeutic ^131^I scan and should not be used to exclude the option of blind ^131^I therapy in patients who are suspected to have metastatic DTC [15].

## 3. Non-Iodine-Related PET Radiopharmaceuticals

### 3.1. ^18^F Tetrafluoroborate ([^18^F]TFB)

Fluorine-18 (^18^F) is a positron-emitting radionuclide produced by a cyclotron. It has a high positron decay ratio (97%), relatively short half-life (109.7 min), and low positron energy (maximum 635 KeV) [24]. The positron energy results in a short diffusion range (<2.4 mm) that favorably increases the resolution limits of the PET images [24]. 

Apart from I^−^, several anions, such as TcO_4_^−^, ReO_4_^−^, ClO_4_^−^, and Br^−^, can be transported via NIS. [^18^F]TFB is a PET isotope labeling with borate anion, used for thyroid and NIS imaging [25,26]. The radiolabeling with fluoride provides a better imaging resolution and a lower effective dose as compared to ^124^I imaging because of its properties [9,25]. It is pharmacologically and radiobiologically safe in humans [25]. The detection rate of local recurrent or metastatic DTC by [^18^F]TFB PET/CT is significantly higher than that of ^131^I diagnostic WBS and SPECT/CT [27]. A pilot study reported that PET/CT imaging with [^18^F]TFB was not inferior to ^124^I in newly diagnosed DTC after total thyroidectomy, and the agreement rate between the two radiopharmaceuticals was 91% [28].

### 3.2. 2-[^18^F]Fluoro-2-deoxy-D-glucose ([^18^F]FDG)

[^18^F]FDG is considered the most well-known radiopharmaceutical in oncologic imaging, commonly synthesized by a nucleophilic substitution reaction [29]. ^18^F replaces the hydroxyl group on the 2-carbon of a glucose molecule [29]. As a glucose analog, it is transported into the cells via glucose transporter (GLUT) protein and phosphorylated into [^18^F]FDG-6-phosphate by hexokinase. However, unlike glucose, [^18^F]FDG-6-phosphate is not further metabolized and consequently trapped within the cells. Therefore, it can be detected by PET imaging [30,31]. Because malignant cells upregulate membrane GLUT proteins (notably GLUT1 and GLUT3) and increase enzymes along the glycolytic pathway, [^18^F]FDG is more accumulated in cancer cells relative to normal cells [30]. 

The “flip-flop” phenomenon between iodine and [^18^F]FDG uptake has been described in DTC. Generally, well-DTC has high iodine and low [^18^F]FDG uptake (Figure 1). In contrast, poorly DTC and anaplastic thyroid cancer have low iodine and high FDG uptake [32,33]. When DTC becomes dedifferentiated, losing the typical morphology and genetic profile of good differentiation, it decreases the iodine concentration, upregulates GLUT, increases [^18^F]FDG uptake, and behaves more aggressively [33,34], as shown in Figure 2.

According to the American Thyroid Association (ATA) guidelines of 2015 for thyroid cancer, [^18^F]FDG PET imaging should be considered in high-risk DTC patients with elevated serum Tg (generally > 10 ng/mL) and negative RAI imaging [1]. Similarly, the Society of Nuclear Medicine and Molecular Imaging and the European Association of Nuclear Medicine recommend performing [^18^F]FDG PET/CT imaging, ultrasound of the neck, and/or CT of the neck, chest, abdomen, and pelvis in DTC patients with elevated Tg and negative diagnostic RAI WBS [10]. There have been controversial reports about the need for TSH stimulation in early [^18^F]FDG PET/CT imaging [35]. A recent meta-analysis reported that the diagnostic performance of PET/CT with TSH stimulation may not be superior to PET/CT without TSH stimulation; however, further well-designed studies evaluating the actual patients’ additional oncological benefit after performing TSH stimulation are necessary [36].

In patients with negative RAI scan and elevated serum Tg, [^18^F]FDG PET/CT had high diagnosis accuracy for detecting recurrent and/or metastatic DTC [36,37]. Caetano et al. reported that the combined sensitivity, specificity, and overall accuracy of [^18^F]FDG PET/CT were 93%, 81%, and 93%, respectively [37], while Qichang et al. found that in a patient-based analysis, the pooled sensitivity, specificity, and receiver operating characteristics curve were 86%, 84%, and 95%, respectively [36]. Vrachimis et al. demonstrated that in patients with DTC and with suspected or known dedifferentiation, [^18^F]FDG PET/MRI was inferior to low-dose [^18^F]FDG PET/CT for the evaluation of pulmonary status, but it was equal to contrast-enhanced neck [^18^F]FDG PET/CT for the evaluation of cervical status. Therefore, the authors suggested obtaining [^18^F]FDG PET/MRI combined with a low-dose CT scan of the chest instead of a high-quality, high-energy, contrast-enhanced CT [38]. In this clinical setting, PET/CT parameters alone or combined with other prognostic factors can help to identify patients with poor outcomes [39]. Albano et al. found that the only independent prognostic factors for overall survival were total metabolic tumor volume and total lesion glycolysis of all lesions with increased [^18^F]FDG uptake [40]. 

Elevated thyroglobulin antibody might indicate recurrent and/or metastatic disease and could be used as an alternative tumor marker for DTC [41]. Several studies showed that [^18^F]FDG PET/CT may be a useful imaging modality to investigate patients with a negative RAI scan and positive/elevated thyroglobulin antibody [42,43,44]. 

An increase in [^18^F]FDG PET/CT imaging leads to increase in thyroid incidentalomas (Figure 3), defined by focal or diffuse [^18^F]FDG uptake in the thyroid gland [9]. A systematic review and meta-analysis of 50 studies reported that the prevalence of [^18^F]FDG-avid focal thyroid incidentaloma was 2.2%, and the risk of malignancy was 30.8%, most of which were PTC [45]. The 2015 ATA guidelines recommended performing fine-needle aspiration of a focal [^18^F]FDG-avid thyroid incidentaloma, sonographically confirmed by a thyroid nodule ≥ 1 cm in size [1]. In contrast, diffuse [^18^F]FDG uptake in the thyroid gland favors benign etiologies. 

Piccardo et al. studied the association between [^18^F]FDG uptake and event-free survival in patients in whom DTC was detected by [^18^F]FDG PET/CT [46]. They concluded that the intense [^18^F]FDG uptake of the primary DTC was associated with the persistence/progression of disease; however, no further prognostic information could be added when all other prognostic factors had been considered [46]. Analysis of imaging by textural features, also known as radiomics, has emerged as an interesting field of research in the past few years. It allows the characterization of the tumor phenotype, which may be achieved by quantitative measurements, each of which is designed to “capture” specific characteristics of an image [47]. Radiomics implies the extraction of features to characterize a volume of interest (VOI) in the images. These features can be categorized into (1) histogram-based features, (2) texture-based features, (3) edge features, and (4) shape features [48]. Gherghe et al. performed a systematic review of the literature on the use of radiomics analysis to discriminate malignant from benign [^18^F]FDG-avid thyroid incidentaloma and demonstrated that the use of PET volumetric parameters and radiomics analysis showed great prospects in the diagnosis and stratification of patients with malignant thyroid nodules [48].

### 3.3. Prostate-Specific Membrane Antigen-Targeting Radiopharmaceuticals

Prostate-specific membrane antigen (PSMA) is a type II transmembrane glycoprotein, expressed not only in prostate cancer but also in the endothelium of tumor-associated neovascular malignancies such as head and neck, breast, bladder, lung, gastric, colorectal, and gynecologic cancers [49]. PSMA-targeting radiopharmaceuticals are typically labeled with Gallium-68 (^68^Ga), such as [^68^Ga]Ga-PSMA-11, [^68^Ga]Ga-PSMA-617, and [^68^Ga]Ga-PSMA I&T, and ^18^F, such as [^18^F]DCFPyL and [^18^F]PSMA-1007 [50,51].

^68^Ga is mainly produced by germanium-68/gallium-68 generator and has a half-life of 68 min. It decays 87.94% through positron emission, with a maximum energy of 1.9 MeV, higher than ^18^F, inherently leading to a lower resolution [51]. In December 2020, [^68^Ga]Ga-PSMA-11, also known as [^68^Ga]Ga-PSMA-HBED-CC, was the first ^68^Ga-labeled radiopharmaceutical approved by the United States Food and Drug Administration (USFDA) for the PET imaging of PSMA-positive prostate cancer [52]. PSMA-11 was developed to enhance the interaction with the PSMA binding site by the initial linking of the acyclic chelator HBED-CC (*N*,*N*′-bis[2-hydroxy-5-(carboxyethyl)benzyl]ethylenediamine-*N*,*N*′-diacetic acid) with the aliphatic spacer 6-aminohexanoic acid (Ahx) and later conjugating with a urea-based PSMA inhibitor to form Glu-NH-CO-NH-Lys(Ahx)-HBED-CC. Radiolabeling was performed by adding ^68^Ga(III) to PSMA-11 [52,53].

Subsequently, [^18^F]DCFPyL, 2-(3-(1-carboxy-5-[(6-[^18^F]fluoropyridine-3-carbonyl)-amino]-pentyl)-ureido)-pentanedioic acid was also approved by the USFDA for the detection of possible early metastatic prostate cancer involvement. It is a second-generation ^18^F-labeled PSMA-targeting tracer and has high tumor:background ratios [54]. 

PSMA was found to be frequently expressed in the microvessels of thyroid tumors but not in benign thyroid tissue. There was heterogeneous PSMA expression in neovasculature ranging from 19% in benign tumors to over 50% in thyroid cancer, and PSMA expression in cancer was significantly higher than that of benign tumors [55]. Ciappuccini et al. examined PSMA expression using immunohistochemistry in 44 DTC patients with neck persistent/recurrent disease [49]. Around 68% of the patients had at least one PSMA-positive lesion, with a similar proportion in RAI-positive and RAI-negative patients. In RAI-negative patients, higher PSMA expression was found in [^18^F]FDG-positive than in [^18^F]FDG-negative patients. Moreover, patients with age ≥ 55 years, primary tumor > 4 cm, or an aggressive subtype had higher PSMA expression, and very high expression was associated with poorer progression-free survival [49]. These findings may lead to new perspectives for the imaging and treatment of DTC, especially in RAI-refractory patients, using PSMA radiopharmaceuticals.

PSMA uptake in DTC on PET/CT imaging has been illustrated [56,57,58,59], as shown in Figure 4.

Metastatic lesions in RAI-refractory DTC patients were demonstrated by PET/CT using ^68^Ga-based PSMA, which found additional metastatic lesions compared with [^18^F]FDG [60,61,62]. Vries et al. treated two RAI-refractory DTC patients with [^177^Lu]Lu-PSMA-617. One showed a modest, temporary response for seven months, while the other experienced disease progression one month after the therapy [62]. The clinical benefits of this therapy in patients with RAI-refractory DTC should be explored.

### 3.4. Somatostatin Receptor-Targeting Radiopharmaceuticals

A radiolabeled somatostatin analog is applicable for imaging tumors with high expression of somatostatin receptor (SSTR), mainly in neuroendocrine tumors (NETs). Currently, radiopharmaceuticals for the PET imaging of SSTR-positive tumors are commonly labeled with ^68^Ga: [^68^Ga]Ga-DOTA-Tyr^3^-octreotate ([^68^Ga]Ga-DOTA-TATE); [^68^Ga]Ga-DOTA-NaI^3^-octreotide ([^68^Ga]Ga-DOTA-NOC); and [^68^Ga]Ga-DOTA-Tyr^3^-octreotide ([^68^Ga]Ga-DOTA-TOC). A macrocyclic bifunctional chelator, DOTA (1,4,7,10-tetraazacyclododecane-1,4,7,10-tetraacetic acid), contains four donor amine nitrogen atoms and four pendant carboxylic acids. It can conjugate small molecules such as somatostatin analogs. DOTA is most commonly used to chelate a wide range of radiometals in various oxidation states, such as Cu(II), In(III), Lu(III), and Ga(III); as a result, stable complexes are formed [63]. The somatostatin analogs exhibit different binding affinities; consequently, the ^68^Ga-labeled somatostatin analogs bind to different SSTR subtypes. [^68^Ga]Ga-DOTA-TATE has the highest affinity towards and selectively binds to SSTR type 2. [^68^Ga]Ga-DOTA-NOC binds to SSTR types 2 and 3, while [^68^Ga]Ga-DOTA-TOC binds to types 2 and 5 [64].

DTC cells show high expression of SSTR types 2, 3, and 5 [65,66]. Therefore, PET/CT imaging with ^68^Ga-labeled somatostatin analogs can possibly detect recurrent or metastatic DTC lesions, especially in patients with RAI-refractory status. In a study, there was a different number and intensity of detected lesions among ^68^Ga-labeled somatostatin analogs. PET/CT imaging with [^68^Ga]Ga-DOTA-TATE found more lesions than with [^68^Ga]Ga-DOTA-NOC and higher lesion uptake in DTC patients with elevated serum Tg and negative post-therapeutic ^131^I scan [67].

Peptide receptor radionuclide therapy (PRRT) using a Lutetium-177 (^177^Lu)- or Yttrium-90 (^90^Y)-labeled somatostatin analog compound is indicated for patients with positive SSTR type 2 expression or metastatic or inoperable NETs [68]. In 2018, [^177^Lu]Lu-DOTA-TATE (Lutathera^®^) was approved by the USFDA for the treatment of positive gastroenteropancreatic NETs. It significantly improved the progression-free survival and response rate among patients with advanced midgut NETs [69]. A meta-analysis study evaluated the therapeutic effect of PRRT in 67 RAI-refractory DTC patients and reported pooled proportions of objective response rate of 15.61%, disease control rate of 53.95%, and serious adverse events of 2.83%. Moreover, PRRT with ^177^Lu had a relatively better therapeutic effect than PRRT with ^90^Y [70]. PRRT may become an alternative treatment for recurrent or metastatic RAI-refractory DTC patients with adequate SSTR expression.

### 3.5. Fibroblast Activation Protein-Targeting Radiopharmaceuticals

Fibroblast activation protein (FAP) is highly expressed in the stroma of several tumor entities. In particular, breast, colonic, and pancreatic carcinomas are characterized by a strong desmoplastic reaction [71].

The FAP expression is generally very low in normal fibroblasts in the human body. In contrast, cancer-associated fibroblasts are distinctively characterized by the overexpression of FAP, having both exopeptidase and endopeptidase activity [71]. Together with extracellular fibrosis, they contribute up to 90% of the gross tumor mass, leaving tumor cells in the minority [71,72]. Therefore, fibroblast activation protein inhibitors (FAPIs) were developed as an anticancer drug and consecutively advanced into tumor-targeting radiopharmaceuticals [72]. A small-molecule inhibitor (UAMC-1110) based on (4-quinoinolyl)glycinyl-2-cyanopyrrolidine scaffold was found to be a highly selective and potent inhibitor of FAP. Various FAPI tracers are 6-quinolyl-modified derivatives of UAMC-1110, such as FAPI-04, FAPI-46, and FAPI-74 [73]. FAPIs linked to ^68^Ga via DOTA chelators are mainly FAPI-04, FAPI-46, FAPI-74, DOTA-2P(FAPI)_2_, DOTA.SA.FAPI, and DATA5m.SA.FAPI [74].

In comparison to [^18^F]FDG, the current standard oncologic PET, the background uptake of ^68^Ga-FAPI tracers in the brain, liver, and oral/pharyngeal mucosa was found to be significantly lower; tumor uptake was not statistically different [71].

The average SUVmax of [^68^Ga]Ga-DOTA-FAPI-04 PET/CT varies among cancers; of those in DTCs, it is not high—the average SUVmax < 6 [72]. A recent prospective clinical trial compared the clinical utility of [^68^Ga]Ga-DOTA-FAPI-04 and [^18^F]FDG PET/CT in 35 metastatic DTC patients and reported significantly higher sensitivity of [^68^Ga]Ga-DOTA-FAPI-04 than [^18^F]FDG for depicting neck lesions (83% vs 65%) and distant metastases (79% vs. 59%) [75]. Chen et al. analyzed [^68^Ga]Ga-DOTA-FAPI-04 PET/CT imaging data of 24 RAI-refractory DTC patients and found that 87.5% of the patients had positive metastatic lesions, with a mean SUVmax of 4.25 [76]. In some cases, ^68^Ga-labeled FAPI showed better detection of metastatic RAI-DTC lesions compared to [^18^F]FDG, owing to the better signal-to-background ratio [77,78].

For therapeutic purposes, Ballal et al. treated 15 metastatic RAI-refractory DTC patients who progressed on sorafenib/lenvatinib and had moderate-to-excellent uptake of [^68^Ga]Ga-DOTA.SA.FAPi, with [^177^Lu]Lu-DOTAGA.(SA.FAPi)_2_. After the treatment, the serum Tg level significantly decreased. Although none of the patients had a complete molecular response, four had a partial response and three had stable disease [79].

## 4. Conclusions

Different PET radiopharmaceuticals have been used for the PET/CT imaging of DTC. ^124^I and [^18^F]TFB are taken up by DTC via NIS. In RAI-refractory disease, the role of [^18^F]FDG PET/CT imaging is well established for metastatic detection, while PSMA, SSTR, and FAP-targeting radiopharmaceuticals are applicable for both imaging and therapy.

## Figures and Tables

**Figure 1 molecules-27-04936-f001:**
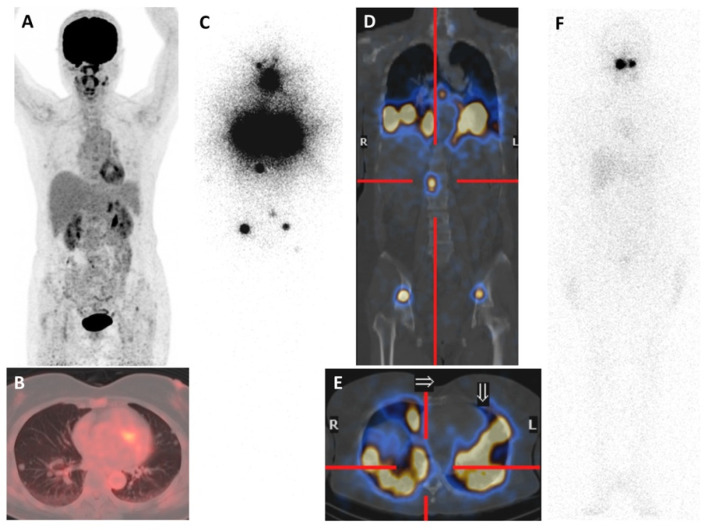
A 59-year-old woman with PTC with lung and bone metastases. She was initially investigated by [^18^F]FDG PET/CT due to multiple lung nodules on chest radiograph. [^18^F]FDG maximum-intensity (MIP) skull-to-mid-thigh PET (**A**) and axial fused PET/CT (**B**) images showed non-FDG-avid small lung nodules. Subsequently, wedge resection of the superior segment of the right lower lung lobe revealed metastatic well-differentiated follicular-derived thyroid carcinoma. After total thyroidectomy, the pre-therapeutic (2 mCi) ^131^I WBS (**C**), coronal (**D**), and axial (**E**) fused SPECT/CT images showed intense iodine uptake in residual thyroid, diffuse lung, and bone metastases. She received ^131^I therapies (cumulative dose of 600 mCi). (**F**) The last post-therapeutic ^131^I WBS image revealed resolution of all iodine-avid lesions, and serum-stimulated Tg declined (<1 ng/mL), implying an excellent response.

**Figure 2 molecules-27-04936-f002:**
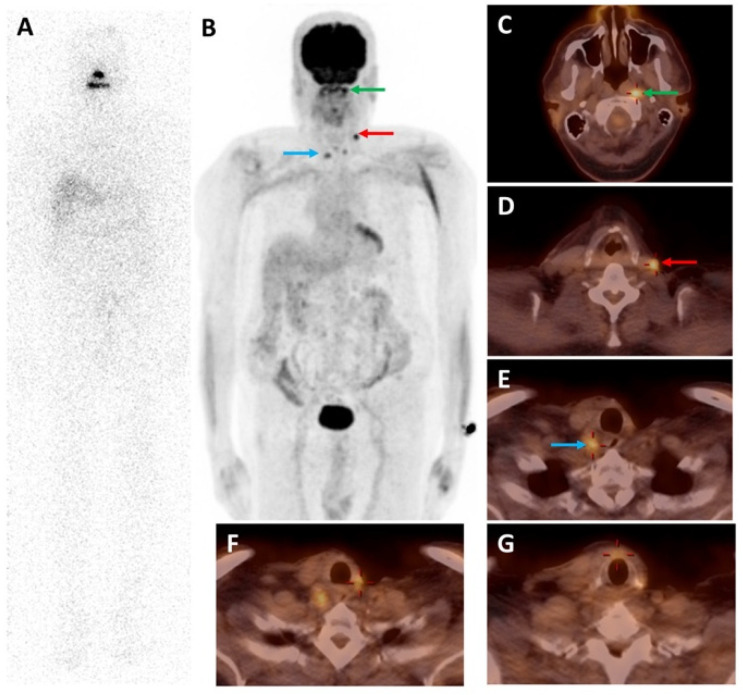
A 70-year-old man with PTC and nodal metastasis was treated with surgery and ^131^I therapy (cumulative dose of 600 mCi). (**A**) The last post-therapeutic ^131^I WBS was negative despite elevated serum Tg. [^18^F]FDG MIP skull-to-mid-thigh PET (**B**) and axial fused PET/CT (**C**–**G**) images showed [^18^F]FDG-avid lesions in left retropharyngeal ((**C**), green arrow), left cervical ((**D**), red arrow), and right cervical ((**E**), blue arrow) lymph nodes, and thyroidectomy bed (**F**,**G**). All are suggestive of RAI-refractory disease.

**Figure 3 molecules-27-04936-f003:**
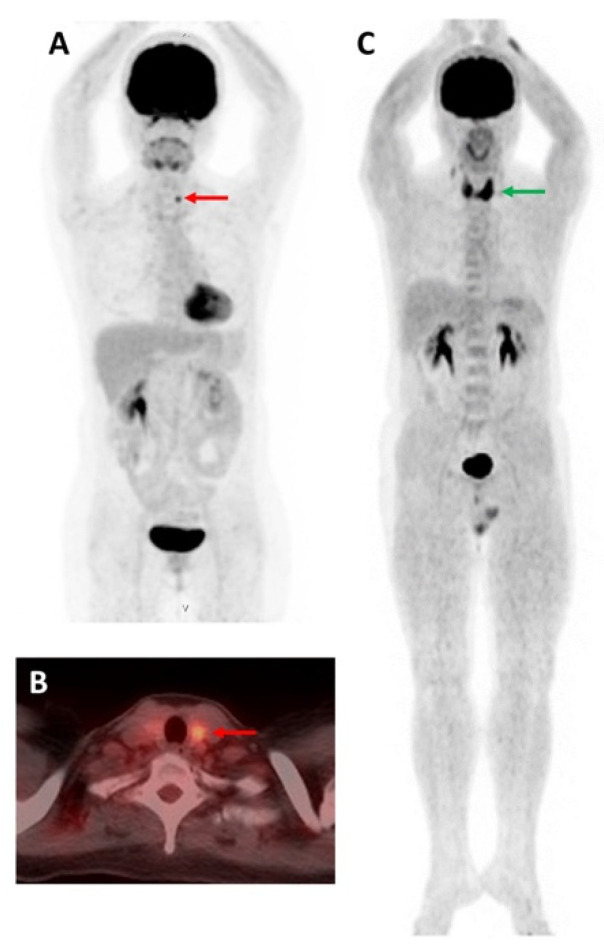
[^18^F]FDG-avid thyroid incidentaloma. (**A**,**B**) A 51-year-old woman with uterine cervical cancer on surveillance. [^18^F]FDG MIP skull-to-mid-thigh PET (**A**) and axial fused PET/CT (**B**) images showed focal [^18^F]FDG uptake in the left thyroid lobe (red arrow). She underwent left thyroid lobectomy and the pathological diagnosis was PTC. (**C**) A 56-year-old man with a fever of unknown origin. [^18^F]FDG MIP whole-body PET image revealed diffuse [^18^F]FDG uptake of the thyroid gland (green arrow). After investigation, he was diagnosed with subacute thyroiditis.

**Figure 4 molecules-27-04936-f004:**
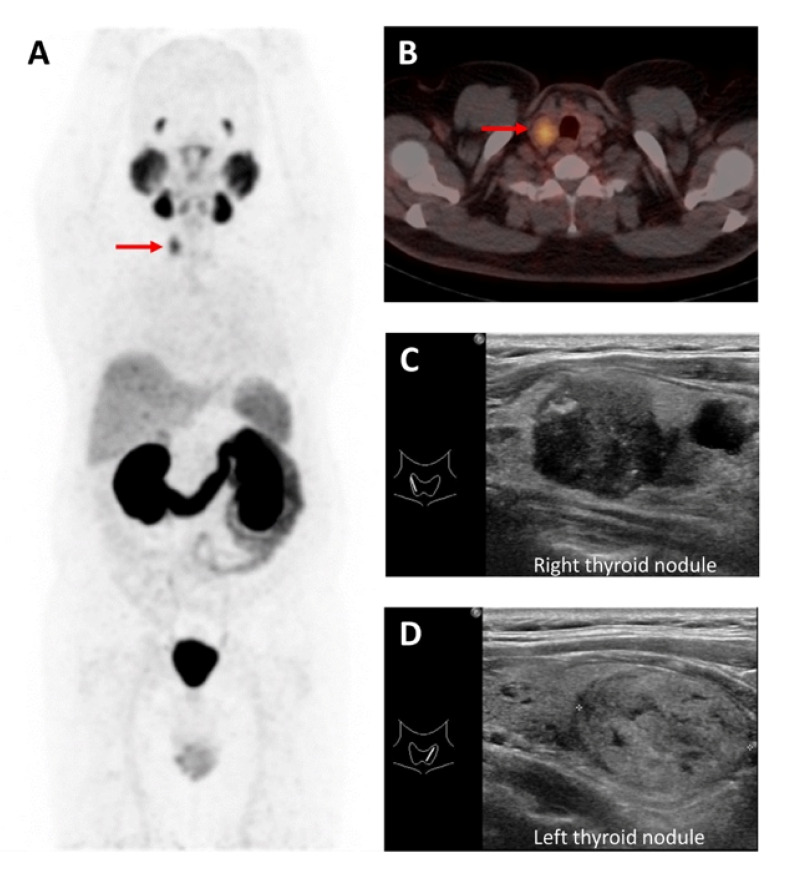
A 59-year-old man with prostate cancer post-radical prostatectomy. [^68^Ga]Ga-PSMA-11 MIP skull-to-mid-thigh PET (**A**) and axial fused PET/CT (**B**) images revealed a PSMA-avid right thyroid nodule (red arrow) despite bilateral thyroid nodules. The ultrasound images (**C**,**D**) revealed right and left thyroid nodules, respectively. He subsequently underwent thyroid surgery with confirmed PTC in the right thyroid nodule and benign left thyroid nodule.

## Data Availability

No new data were created or analyzed in this study. Data sharing is not applicable to this review article.

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
