# Peer review of "Positron Emission Tomography Radiopharmaceuticals in Differentiated Thyroid Cancer"

_molecules, 2022, doi:10.3390/molecules27154936_

Round 1

Reviewer 1 Report

Dear authors, 

this is a well performed review about the role of positron imaging in DTC.
Some issues with english language are present, therefore I suggest to revise the article and improve the level of the writing.
Furthermore, I have some comments about the section of 18F-FDG imaging. First of all, you underlined the possible role of PET/CT for the assessment of thyroid incidentalomas and in this setting I believe that a brief discussion on radiomics is necessary. Similarly, another discussion has to be made about the prognostic role of 18F-FDG PET/CT in DTC.  

Author Response

Dear Reviewer,

Thank you for your comments on the manuscript title “Positron Emission Tomography Radiopharmaceuticals in Differentiated Thyroid Cancer”. We have revised our manuscript accordingly.

Reviewer’s comments:

  1. Discussion on radiomics in [18F]FDG-avid thyroid incidentaloma

Response: Information about radiomic analysis is added in [18F]FDG section.

  1. Discussion about prognostic role of [18F]FDG PET/CT in DTC

Response: Information about prognostic value of [18F]FDG PET/CT in radioiodine-refractory DTC and [18F]FDG-avid thyroid incidentaloma  is added in [18F]FDG section.

Best regards,

Chaninart Sakulpisuti

Reviewer 2 Report

This review article highlights imaging by PET radiopharmaceuticals focusing on molecular pathways and applications in differentiated thyroid cancer (DTC) which has been practically performed by radioiodines such as iodine-123 and iodine-131. Since DTC is very well diagnosed by radioiodine imaging, various studies have not been reported. Nevertheless, this review contains sufficient data to diversify the knowledge of researchers in related fields. Appropriate references are cited in the paper. It seems, however, that some modifications are needed in terms of format to journal guidelines.

In addition, the reviewer recommends two minor modifications.

1. “Section 2.2. 18F Tetrafluoroborate ([18F]TFB)” seems appropriate to move to the “Non-iodine-related PET radiopharmaceuticals” section.

2. If the chemical structure of the PET radiopharmaceutical mentioned in the manuscript is added, it will fit well with the direction of the journal and the readers.

Author Response

Dear Reviewer,

Thank you for your comments on the manuscript title “Positron Emission Tomography Radiopharmaceuticals in Differentiated Thyroid Cancer”. We have revised our manuscript accordingly.

Reviewer’s comments:

  1. Section “2.2 18F Tetrafluoroborate ([18F]TFB)” seems appropriate to move to the “Non-iodine-related PET radiopharmaceuticals” section.

Response: It is moved to “Non-iodine-related PET radiopharmaceuticals” section as “3.1 18F Tetrafluoroborate ([18F]TFB” section.

  1. If the chemical structure of the PET radiopharmaceuticals mentioned in the manuscript is added, it will fit well with the direction of the journal and the readers

Response: Information about the general chemical structure of mentioned PET radiopharmaceuticals is added in [18F]FDG, PSMA-, SSTR-, and FAP- targeting radiopharmaceuticals section.

Best regards,

Chaninart Sakulpisuti